# The Domain and Microstructure of Resin-Bonded Magnets

**DOI:** 10.3390/ma14112849

**Published:** 2021-05-26

**Authors:** Marcin Jan Dośpiał

**Affiliations:** Faculty of Production Engineering and Materials Technology, Czestochowa University of Technology, 19 Armii Krajowej Av., 42–200 Czestochowa, Poland; mdospial@wp.pl

**Keywords:** bonded permanent magnets, X-ray diffractometry, domain structure, microstructure, Mössbauer spectroscopy, transmission electron microscopy, magnetic force microscopy

## Abstract

This paper presents domain and structure studies of bonded magnets made from nanocrystalline Nd-(Fe, Co)-B powder. The structure studies were investigated using scanning electron microscopy (SEM), high-resolution transmission electron microscopy (HRTEM), Mössbauer spectroscopy and X-ray diffractometry. On the basis of performed qualitative and quantitative phase composition studies, it was found that investigated alloy was mainly composed of Nd_2_(Fe-Co)_14_B hard magnetic phase (98 vol%) and a small amount of Nd_1.1_Fe_4_B_4_ paramagnetic phase (2 vol%). The best fit of grain size distribution was achieved for the lognormal function. The mean grain size determined from transmission electron microscopy (TEM) images on the basis of grain size distribution and diffraction pattern using the Bragg equation was about ≈130 nm. HRTEM images showed that over-stoichiometric Nd was mainly distributed on the grain boundaries as a thin amorphous border of 2 nm in width. The domain structure was investigated using a scanning electron microscope and metallographic light microscope, respectively, by Bitter and Kerr methods, and by magnetic force microscopy. Domain structure studies revealed that the observed domain structure had a labyrinth shape, which is typically observed in magnets, where strong exchange interactions between grains are present. The analysis of the domain structure in different states of magnetization revealed the dynamics of the reversal magnetization process.

## 1. Introduction

In the coming years, due to the development of 3-D printing technology and the creation of the first magnetic filaments [1], the interest in bonded magnets, their research techniques and the design of magnetic properties is expected to increase.

Due to the reduced content of magnetic material in a volume of the bonded magnet, from the powders used to their production, the best magnetic properties are required, i.e., the greatest possible value of magnetic energy product (BH)_max_, coercivity _j_H_c_, remanence M_r_ and getting closest to one M_r_/M_s_ ratio. The formation of such properties in powders based on the rare earth (RE)-Fe-B compound (where, RE—rare earth) is associated with the generation of the relevant structure and proper phase composition [2,3,4,5,6,7,8,9,10,11]. 

Modifications of phase composition, which relate to the proportion of basic elements used during production, can involve designing the alloy composition based on stoichiometric RE_2_Fe_14_B, over-stoichiometric transition metal (TM), or over-stoichiometric rare earth (RE) elements [2,3,4]. 

An over-stoichiometric TM or RE addition results in the formation of a composite structure. Extra TM content is used in order to obtain material composed of two kinds of grains representing soft and hard phases. However, while simply introducing an addition will not achieve the desired magnetic properties, it is also necessary to generate a fine microstructure. Such fragmentation of the structure allows the formation of exchangeable interaction between grains of both phases. The presence of the soft phase coupled with the hard magnetic phase is desired in view of improving remanence and saturation of magnetization but simultaneously leads to deterioration of coercivity strength [2,3,4,5,6].

In the case of RE-Fe-B magnets with extra RE content, the proper selection of manufacturing parameters can lead to the formation of a structure composed of an amorphous, paramagnetic phase with a thickness of several nm and the insulation of grains of the hard magnetic phase. Such a structure simultaneously increases the potential barrier in the reversal magnetization process and keeps the exchangeable interaction between adjacent grains. This, in turn, guarantees the improvement of coercivity strength, with simultaneous, small deterioration of remanence and saturation of magnetization in comparison to stoichiometric magnets with a fine structure [7,8].

The visualization of the described structures is shown in Figure 1.

Other types of modifications rely on the partial exchange in the type of RE [9,10] or TM elements [11,12]. Such modifications are associated with substitutions of atoms in the unit cell. 

In this paper, the micro and domain structure of an Nd-Fe-B-bonded magnet with over-stoichiometric Nd content was analyzed. Both structures were observed through the application of various complementing methods. Additionally, the domain structure was studied, in the case of reversal magnetization, by observing the studied sample at different remanent states. 

## 2. Materials and Methods

### 2.1. Materials

The studied magnets were obtained from an isotropic powder, obtained from quenched Nd_14.8_Fe_76_Co_4.8_B_4.2_Nb_0.2_ ribbons (Magnequench, Chuzhou, China), by compression molding at a pressure of 900 MPa. Epoxy resin (2.5 wt%, Tele and Radio Research Institute, Warsaw, Poland) was used as a binder and zirconium stearate resin (0.5 wt%, Sigma-Aldrich Sp. z o.o., Poznan, Poland) as a lubricant. Samples were cured at 180 °C for 2 h.

### 2.2. Phase Composition and Microstructure Studies

In order to specify the qualitative phase composition of the investigated magnets, X-ray diffractometry, Mössbauer spectroscopy, transmission and scanning electron microscopy studies were performed. 

X-ray diffraction patterns were taken using the Bruker D8 ADVANCE X-ray diffractometer (XRD, Bruker, Billerica, MA, USA) equipped with a semi-conductor counter. Diffraction patterns were made using a Cu K-α radiation source (1.541 Å) in Bragg–Brentano geometry. Samples for X-ray measurements were powdered by low-energy milling, then scanned in 2θ range from 30 to 120° with an angle step of 0.02° and an exposure time of 3 s. The average grain size was estimated from 10 Bragg’s reflections with the highest intensities using the Bragg equation.

The Mössbauer studies were carried out using a Polon 2330 Mössbauer spectrometer (POLON, Kraków, Poland) with a constant acceleration γ-ray source ^57^Co, with a 50mCi activity, mounted in a rhodium matrix at room temperature. The spectrometer was calibrated using a <-Fe thin film. The spectra were analyzed using the NORMOS evaluation program.

Microstructure observations were performed using a ZEISS Neophot 32 metallographic light microscope (Carl Zeiss Industrielle Messtechnik GmbH, Oberkochen, Germany), and a Vega 5135 MM scanning electron microscope (SEM, TESCAN, Brno–Kohoutovice, Czech Republic). The surfaces of the samples for microstructure observations were prepared using standard metallographic procedures: wiping samples on abrasive paper and polishing them on diamond pastes of different gradations, i.e., 7–10, 2 and 0.5 μm, respectively.

Studies performed using JEM 3010 high-resolution transmission electron microscope (JEOL USA Inc., Peabody, MA, USA) were carried out. The resolving power of the JEM 3010 electron microscope was 0.17 nm, and the accelerating voltage was 300 kV. The microscope was equipped with a camera, which allowed the recording of microscopic images in a digital format. The chemical composition was analyzed using energy-dispersive X-ray spectroscopy analysis (EDS). The sample for the transmission electron microscopy studies was in the form of thin films obtained by the initial wiping of the sample on abrasive papers of different gradations, followed by ion thinning using a Gatan PIPS 691 ion polisher(Gatan, Las Positas Blvd. Pleasanton, CA, USA).

### 2.3. Domain Structure Studies

The study of the domain structure was carried out using a ZEISS Neophot 32 metallographic light microscope, allowing the observation of the domain structure via the Kerr and Bitter methods, a Vega 5135 MM scanning electron microscope (SEM) via the Bitter method and a Nanoscope magnetic force microscope (MFM).

Samples for domain structure observations were prepared using standard metallographic procedures. For observations through the use of a scanning electron microscope, Fe_3_O_4_ colloidal suspension with particle size about 10 nm was applied to the surface of the samples. After drying the colloidal suspension on the surface of the sample, a thin copper layer was deposited.

The study of the domain structure using MFM was completed with magnetic contrast imaging as performed in the tapping (Atomic Force Microscopy)/lift (Magnetic Force Microscopy) mode. The signal was obtained by measuring the phase shift of a cantilever oscillating at a resonant frequency. An MESP-ESP tip coated with CoCr films at a thickness of ≈35 nm, with a coercivity of 32 kA/m and at a scan height of 100 nm, was used.

## 3. Results and Discussion

### 3.1. Phase Composition Studies

Figure 2 presents the X-ray diffraction pattern of the studied alloy. The comparison of the X-ray pattern with the PDF-2 database revealed that all peaks of most intensities were characteristic of the Nd_2_Fe_14_B phase. 

The X-ray patterns were used to evaluate the moderate grain sizes from Bragg’s reflections with the highest intensities using the Equation (1) [13]
(1)$$\Delta^{\mathrm{hkl}}\left(2\theta \right)\cdot \cos\left(\theta_{B}^{\mathrm{hkl}}\right)=\frac{k\cdot \lambda }{D}+2\frac{\Delta d}{d}\cdot \sin\left(\theta_{B}^{\mathrm{hkl}}\right),$$
where D—grain size, k—shape factor equal to 0.89, λ—X-ray wavelength, θ^hkl^ is the line full width at a half maximum intensity (FWHM) in radians, Δd/d—relative lattice strain and is Bragg’s angle. The presented dependence (1) has a linear character, and the grain size can be calculated from the intersection of the straight line with the ordinate. The relation (1) considers the effect of the grain size and strain on the half widths of the diffraction peaks. Due to the lack of reference samples, the apparatus factors were not eliminated, and the peak broadening caused by the strain was not taken into account. The moderate grain size was equal to 132 nm.

Basing on the obtained results, it can be stated that, due to the complex chemical composition of the studied Nd_14.8_Fe_76_Co_4.8_B_4.2_Nb_0.2_ alloy, the presence of small amounts of different phases in the volume of the sample could not be excluded. A large number of peaks characteristic of the Nd_2_Fe_14_B phase and the fine–moderate grain size of the main phase (≈130 nm) could result in overlapping and broadening of peaks resulting from constituent phases on the XRD pattern, respectively. This leads to problems with phase composition identifications, especially for phases with a volume content lower than 5%. Therefore, the phase composition was additionally confirmed by Mössbauer studies.

The Mössbauer studies were used to analyze the qualitative and quantitative phase composition of the studied sample.

Due to the polycrystalline nature of the powders and the random orientation of the magnetization vectors in relation to the γ flux quantum, it was assumed that the ratio of the relative intensity of the lines in the sextet was 3:2:1:1:2:3, and absorption lines were Lorentzian lines of equal width. It was also assumed that γ photon absorption for all positions of iron atoms had equal probability.

Figure 3 shows the Mössbauer spectra of the investigated samples. The measured spectrum was decomposed into six independent Zeeman sextets corresponding to six non-equivalent positions of iron in the elementary Nd_2_Fe_14_B cell and a paramagnetic doublet, which was attributed to the Nd_1.1_Fe_4_B_4_ paramagnetic phase.

In Figure 3, the experimental points are marked by squares, and the best fit, as well as constituent sextets, are marked by solid lines. Table 1 presents the values of hyperfine parameters, such as the isomeric shift, quadrupole splitting and induction of hyperfine field. Each of the sextets was assigned to positions corresponding with the proper iron crystallographic positions, 4e, 4c, 8j_1_, 8j_2_, 16k_1_, 16k_2_ (stored in ascending order of intensity), on the basis of the dependence of line intensities and the number of occupations of these positions by iron atoms.

It was assumed that the hyperfine field in the i-position was directly proportional to the magnetic moment of iron at this position
(2)$$\mu_{0}\cdot H_{\mathrm{hyp}}^{i}=A\cdot \mu_{\mathrm{Fe}}^{i}$$
where A was a constant of proportionality equal to
(3)$$\left(14-15\right)\frac{T}{\mu_{B}}$$
which was determined experimentally [14,15].

In the assignment of the sextets to positions k_1_ and k_2_, it was taken into account that, in the immediate vicinity of the k_1_ position, one boron atom was present, which reduces the magnetic moment of iron. In the immediate vicinity of the position k_2_, there was no boron; therefore, the hyperfine field at this position was increased. 

In the vicinity of position e, there were two boron atoms, and in position c, there were no boron atoms. Therefore, in position e, there was a lower hyperfine field.

In the studied Nd-Fe-B type of powder, 4.9% of iron atoms were replaced by cobalt. A study by E.T. Dwayne et al. [16] shows that the cobalt atoms do not occupy different crystallographic positions in a statistical manner, avoiding the particular location j_2_ and preferring positions j_1_ and k_2_. This reduces the intensity of the line of speed 6 mm/s up to its disappearance, with the increase of the percentage share of the cobalt atoms (30% Co [16]). This was due to a lower number of iron atoms in the vicinity of the Fe atom in the position j_2_, because they were replaced by cobalt. Moreover, 4.9%. of cobalt in the sample does not significantly affect the Mössbauer spectrum.

The phase compositions obtained on the basis of fitting parameters of Mössbauer spectra are shown in Table 2.

It can be concluded that the performed analysis confirms the results of Mössbauer spectroscopy studies. As such, the best fit of the hyperfine field for different non-equivalent Fe sites in an Nd_2_Fe_14_B elementary cell is achieved for $H_{\mathrm{hyp}}\left(8j_{2}\right)>H_{\mathrm{hyp}}\left(16k_{2}\right)>H_{\mathrm{hyp}}\left(16k_{1}\right)>H_{\mathrm{hyp}}\left(8j_{1}\right)>H_{\mathrm{hyp}}\left(4c\right)>H_{\mathrm{hyp}}\left(4e\right)$ [17,18,19].

### 3.2. Structure Studies

Figure 4 presents photos of the surface of the investigated magnet, captured using a Neophot 32 metallographic light microscope and a Vega 5135 MM scanning electron microscope. The photos have been selected to show both large and small particles. The particle size was determined as the moderate length from the weight point.

Longitudinal particles with a width of up to 40 μm are visible in the photos, which are showing accurately crushed ribbons and larger particles of up to 100 μm, representing less thoroughly crushed ribbons. The dark areas between the particles come from the presence of the binder. Based on 40 photos, including those presented in Figure 4, the distribution and mean value of the particle size were determined. The distribution was obtained based on measurements of 800 particles. The results are shown in Figure 5.

The best fit was obtained for the lognormal distribution. The probability density function for this distribution is expressed by the formula [20]:(4)$$p\left(\delta \right)\cdot d\delta =N\cdot \frac{1}{\delta }\cdot e^{\frac{-{\left[\log\delta -\mu \right]}^{2}}{2\sigma^{2}}},$$
where μ is associated with the mean value of particle δ_av_ by the following formula: (5)$$\delta_{\mathrm{av}}=\mu +\frac{\sigma^{2}}{2\cdot \log\left(e\right)}$$
while N was evaluated from normalization condition:(6)$$\int_{1}^{\sigma_{\max}}p\left(\delta \right)\cdot d\delta =1,$$

The largest occurrence probability has particles with a diameter of 7 μm. 

Figure 6 presents an exemplary transmission electron microscope image.

Polyhedral grains of different sizes are visible in the image (Figure 6). The grain size distribution evaluated on the basis of TEM images (with a lower scale) is presented in Figure 7. The statistics were made on the basis of 850 grains. 

Lognormal distribution was also used for fitting. It was found that the moderate grain size was equal to 163 nm and the largest probability of occurrence had particles with a diameter of 127 nm.

Further analysis of TEM images showed (Figure 8) the presence of some crystallographic inclusions in the structure of the examined sample. 

The chemical composition analysis for the 2acb area showed Fe at 65.7%., Co at 19.1%, Nd at 14.8% and trace amounts of La at 0.4%. A chemical composition analyzer, coupled with an electron microscope, does not detect the presence of boron; therefore, according to the literature, it was assumed that the boron content is 5.8% [21]. Analysis of the examined electronograms corresponding to presented areas showed that the 2acb area corresponds to the grain of the Nd_2_(Fe, Co)_14_B phase, which was also confirmed by an analysis of the reflexes on the electronograms. 

Designated d_hkl_ spacing between the planes in the atomic lattice is summarized in Table 3 and compared with the theoretical data. Slight differences in distances d_hkl_ are the result of replacing the iron atoms with cobalt.

The analysis of the 2aca area showed Fe at 14.5%, Co at 5% and Nd at 79.7%, and that the area 2aca corresponds to Nd-rich inclusion.

The main Nd_2_Fe_14_B-phase grains show no signs of structural defects. Precipitations of the other phases are seen at grain boundaries. They provide additional, strong pinning sites of domain walls in studied ferromagnetic material.

Figure 9 shows high-resolution TEM images. Figure 9a (inset) shows four distinguished areas: A1–A4. The first three represent the same type of structure, which can be inferred on the basis of an 0.88 nm distance between parallel lines. This distance represents the basal plane of the tetragonal Nd_2_Fe_14_B elementary cell [22].

The structure of small crystallographic inclusion, marked on the figure as area A4, represents the grain of α Fe phase. The measured distance between crystallographic planes is equal to 0.2 nm and represents (1 1 0) a plane of a body-centered cubic (bcc) crystal structure of α-Fe phase [22]. Inclusions of the other phases visible at grain boundaries are described in the literature as additional domain wall pinning centers [23]. Additionally, in Figure 9a (inset), the area representing an amorphous Nd-rich region with a thickness of about 2 nm is highlighted. The thickness of paramagnetic isolation of Nd_2_Fe_14_B grains was measured based on the photo presented in Figure 9b, clearly distinguishable from the background. The presence of this layer, at the grain boundaries, contributed the growth of coercivity and a reduction in the remanence of the permanent magnet [24].

Despite the fact that the X-ray diffraction and Mössbauer spectroscopy did not reveal the presence of an amorphous phase in the sample volume, Figure 9c shows partially crystallized areas (A1–A3) in an amorphous matrix. Figure 9d shows two neighboring areas, i.e., a crystallized Nd_2_Fe_14_B grain and an amorphous grain, obtained by high-resolution TEM.

### 3.3. The Domain Structure Studies

Figure 10 shows the domain structure obtained by the magneto-optical Kerr effect.

Visible areas of various shades of gray (contrast) are the magnetic domains showing different directions of magnetization vectors. Polarized light falling on the surface of the polished sample changes the polarization plane under the influence of the magnetic moments present on the surface of a magnet. Domains are observed when the magnetic charge density on the surface is low or zero; therefore, by using this method, we observed areas of domains and not their boundaries [25]. The diameter of observed domains is in a range between 0.5 and 10 μm.

Figure 11 shows the domain structure obtained by Bitter powder figures, using a metallographic microscope. Dark areas on the particle of Nd_14.8_Fe_76_Co_4.8_B_4.2_Nb_0.2_ powder originate from magnetic ferrofluid particles grouped together and represent the boundaries between the domains. The white areas represent the magnetic domains [26].

The observed domain structure has a labyrinth shape [27]. This kind of domain structure is typically observed in magnets where strong exchange interactions between grains are present [26,27]. The suspension of the colloidal solution is concentrated in areas where a large change in the magnetic field on the sample surface is present. In the case of the magnets in which the interaction domains are present, the magnetic moments of neighboring grains, which significantly differ in the direction of magnetization, lead to the formation of the magnetic poles on the surface and thus cause the clustering of ferrofluid in the grain boundaries.

SEM images of the studied sample at different remanent states are presented in Figure 12a–h.

Figure 12a,b shows images of the domain structure for the non-magnetized sample, made from the same area, magnified 6000 and 12,000 times, respectively. The observed structure is the same labyrinth structure as shown in Figure 11, only at a higher magnification. The domain structure in the remanence state after the external magnetic field was magnetized by 0.6 T is shown in Figure 12c. The observed domains are in the form of chains oriented parallel to the applied magnetic field. The applied 0.6 T field does not exceed the field value needed for pinning the domains, which only assists in their orientation. A similar structure is observed over the entire surface of the sample. Figure 12d–g shows the domain structure of four different areas of the sample, in the remanence state, after magnetizing the external magnetization field by 0.9 T, a value close to coercivity of the sample. 

In Figure 12d, it can be seen that chain domains are present only in part of image, while in Figure 12e,f they occupy the entire image but have different sizes. In Figure 12g, only parts of domains, which are oriented alongside the applied external magnetic field, are visible. This shows the distribution of the pinning sites in different parts of the sample. The visible area in Figure 12h represents the sample in the state of remanence after magnetizing the external field by a value of 1.2 T. In this image, the domain structure is no longer observed. There are almost no domains over the entire surface, although there are still some parts of their leftovers. A method for reducing the domains indicates that the applied magnetic field causes the reversal magnetization of grains located on the borders of the interaction domain [28].

The domain structure in different remanent states has also been studied by the use of magnetic force microscopy. An exemplary MFM image is shown in Figure 13. Visible domain structures for sample in the remanence state, after magnetization by 0.6 T, is similar as that shown in Figure 12c. The observed domains are in the form of chains oriented parallel to the applied magnetic field. The linear analysis of magnetic field fluctuations over the sample revealed that the observed image consists of a superposition of two types of oscillations. The first represents a diameter of domains and varies from 0.5 to 10 μm (from 0.5 to 2.5 μm on presented image). The second is similar to the diameter of Nd-Fe-B grains (around 140 nm), determined from structure studies.

## 4. Conclusions

On the basis of X-ray diffraction analysis and Mössbauer studies it was found that Nd_14.8_Fe_76_Co_4.8_B_4.2_Nb_0.2_ magnetic powder consists of about 98% of Nd_2_(Fe, Co)_14_B, 2% of Nd_1.1_Fe_4_B_4_ and trace amounts of other phases.

Based on the transmission electron microscope images, the grains of the main phase were identified as Nd_2_(Fe, Co)_14_B. Observed grains statistically did not reveal structural defects and were surrounded by a 2–4 nm wide amorphous insulation layer. The presence of this layer at the grain boundaries contributed to the growth of coercivity (the layer is the pinning center of domain wall) and the reduction of remanence (what results from the increase in the distance between adjacent grains and the weakening of the exchange interactions) of the permanent magnet, which is in agreement with results obtained by Hossein Sepehri-Amin et al. [24]. The performed phase studies did not permit the determination of the quantitative value of paramagnetic amorphous phase content, consisting of an excess of Nd. Inclusions of the other phases were visible at the grain boundaries and constituted additional, strong domain wall pinning centers; similar behavior was described in [23]. The microstructure studies have also allowed us to determine the distribution of Nd_14.8_Fe_76_Co_4.8_B_4.2_Nb_0.2_ powder particle size as well as grain sizes. The moderate grain size was smaller than the single-domain particle size for the Nd-(Fe,Co)-B phase [29]. Therefore, the expected domain structure in the studied material was the labyrinth-like domain structure controlled by exchange interactions.

The domain structure observations using the magneto-optic Kerr effect allowed us to determine the domain sizes, which varied in a range between 0.5 and 10 μm. These results were also confirmed by the linear analysis of magnetic field fluctuations (large ones) on MFM images. Both results were coherent. The observations of domain structure by the Bitter powder figures method revealed a labyrinth-like type of domain structure. Similar structures were also observed on MFM images. Additionally, the linear analysis of MFM images revealed, next to large fluctuations of the magnetic field, the existence of small ones. Their size was similar to the grain size.

The dynamics of domain structure changes was illustrated by its analysis in different remanent states, using scanning electron microscopy and the Bitter powder figures method. It was found that the sample magnetized by field value lesser than its coercivity (0.6T) had domains in the form of chains oriented parallel to the applied magnetic field. Magnetization using the field of value close to the coercivity (0.9T) led to a state where chain domains were present only on part of the image and were smaller in size. Further magnetization (1.2T) led to the vanishing of domains over the entire surface, although there were still some parts leftover. The obtained results indicate that the applied magnetic field caused the reverse magnetization of grains located on the borders of the interaction domains [26,28].

## Figures and Tables

**Figure 1 materials-14-02849-f001:**
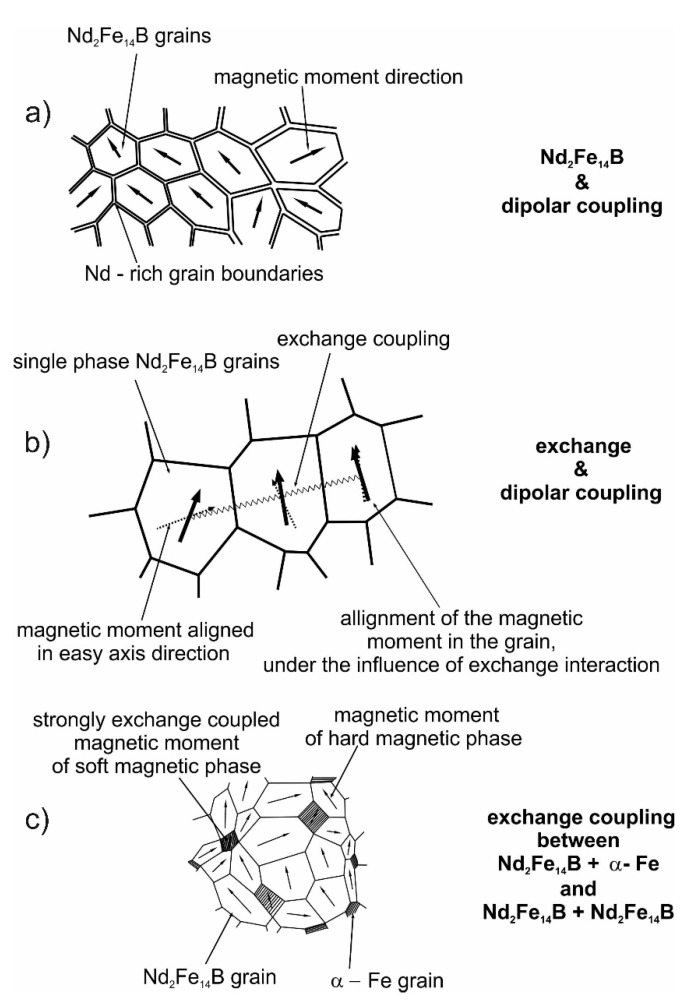
Three types of structures of magnets obtained by different designs of alloy composition based on (**a**) over-stoichiometric rare earth (RE) element, (**b**) stoichiometric RE_2_Fe_14_B and (**c**) over-stoichiometric transition metal (TM).

**Figure 2 materials-14-02849-f002:**
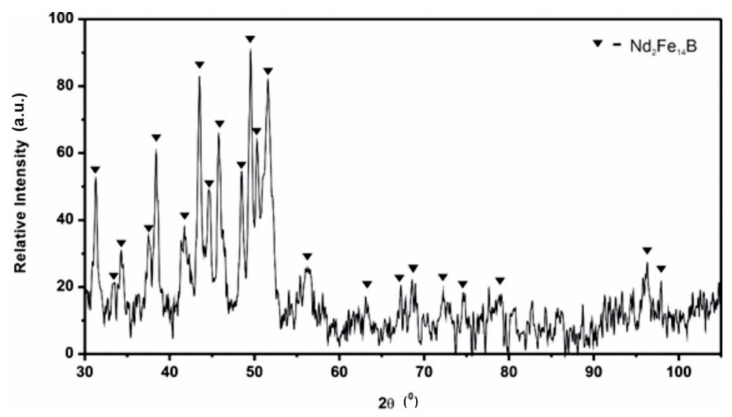
X-ray diffraction patterns for Nd_14.8_Fe_76_Co_4.8_B_4.2_Nb_0.2_ powder.

**Figure 3 materials-14-02849-f003:**
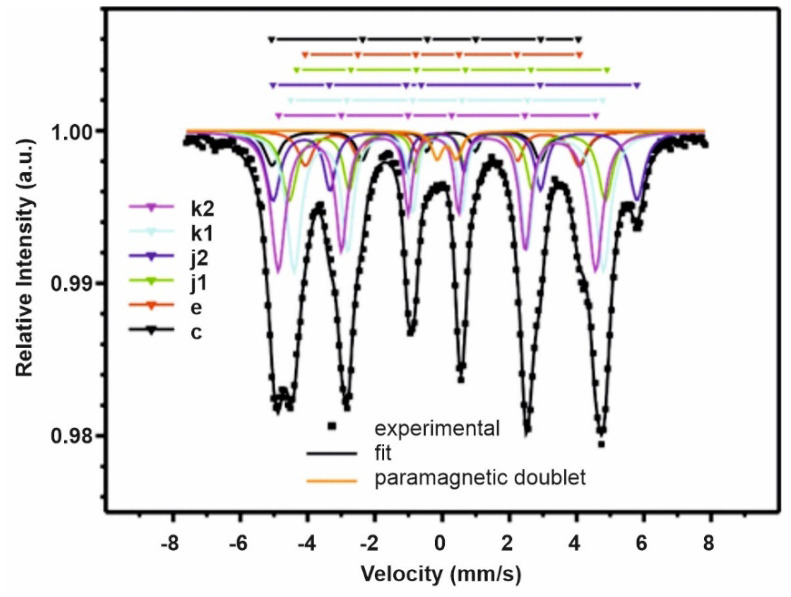
Mössbauer spectrum of bonded magnet prepared from Nd_14.8_Fe_76_Co_4.8_B_4.2_Nb_0.2_ powder. At the top of the picture, the positions of the lines corresponding to the constituent Zeeman sextets for various Fe crystallographic positions are provided.

**Figure 4 materials-14-02849-f004:**
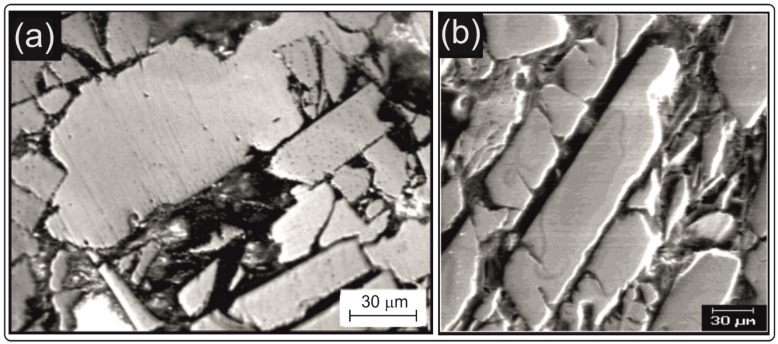
The microstructure of two different areas of the sample, obtained using (**a**) the Neophot 32 metallographic microscope and (**b**) Vega 5135 MM scanning electron microscope.

**Figure 5 materials-14-02849-f005:**
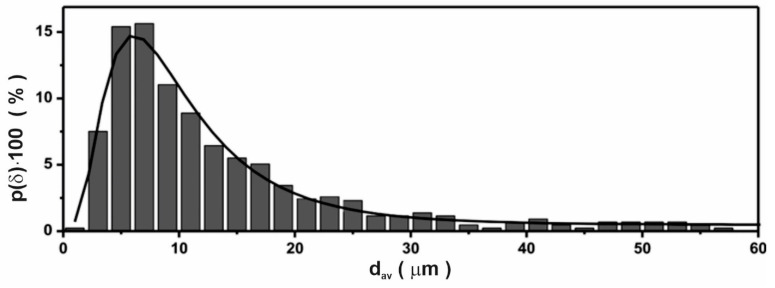
The lognormal distribution of particle diameter obtained on the basis of microstructure images.

**Figure 6 materials-14-02849-f006:**
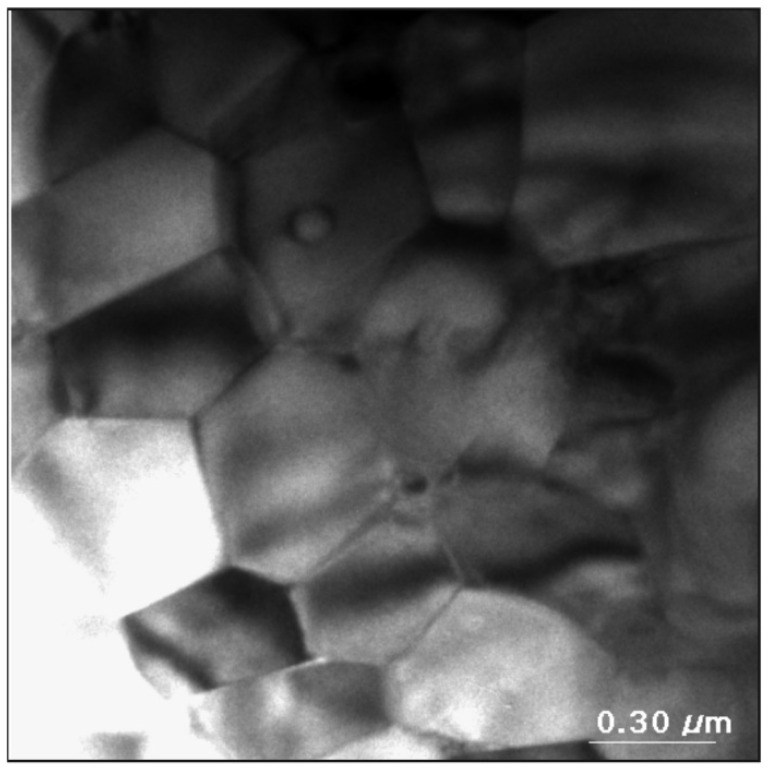
The grain microstructure of a Nd_14.8_Fe_76_Co_4.8_B_4.2_Nb_0.2_ bonded magnet obtained by a transmission electron microscope.

**Figure 7 materials-14-02849-f007:**
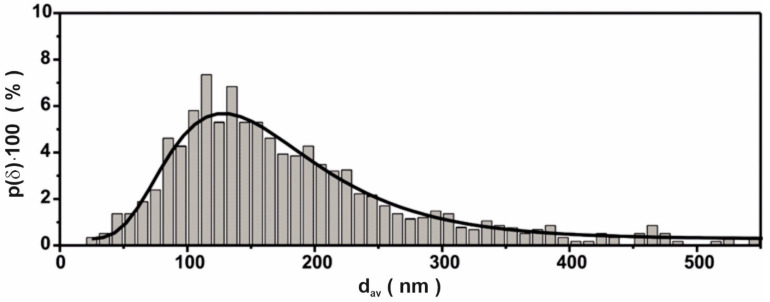
The lognormal distribution of grain size diameter obtained on the basis of TEM images.

**Figure 8 materials-14-02849-f008:**
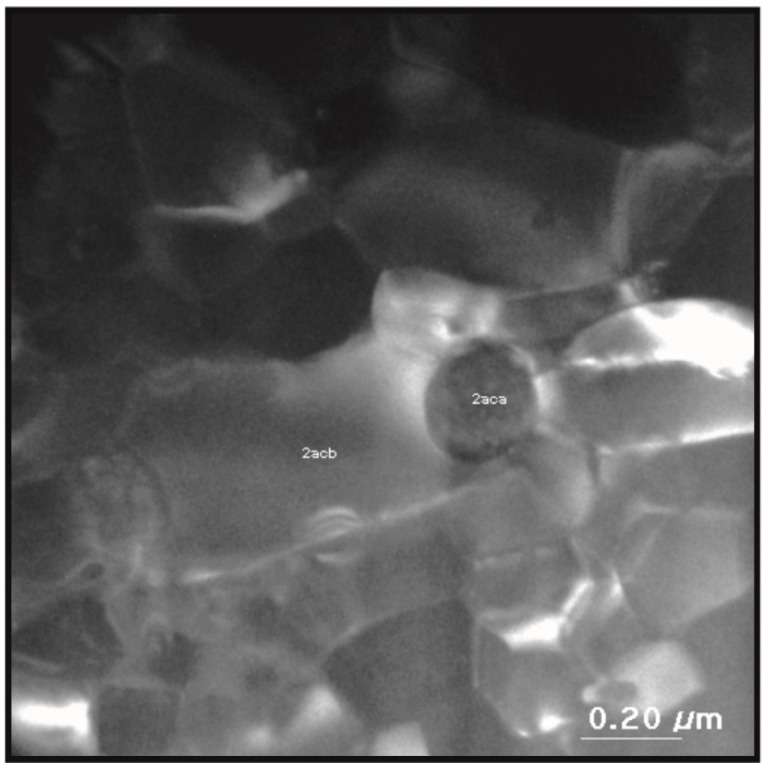
The microstructure of the bonded magnet based on the Nd_14.8_Fe_76_Co_4.8_B_4.2_Nb_0.2_ compound, obtained by a transmission electron microscope with marked areas corresponding to inclusion of Nd-rich phase (2aca) and the Nd-(Fe, Co)-B grain (2acb).

**Figure 9 materials-14-02849-f009:**
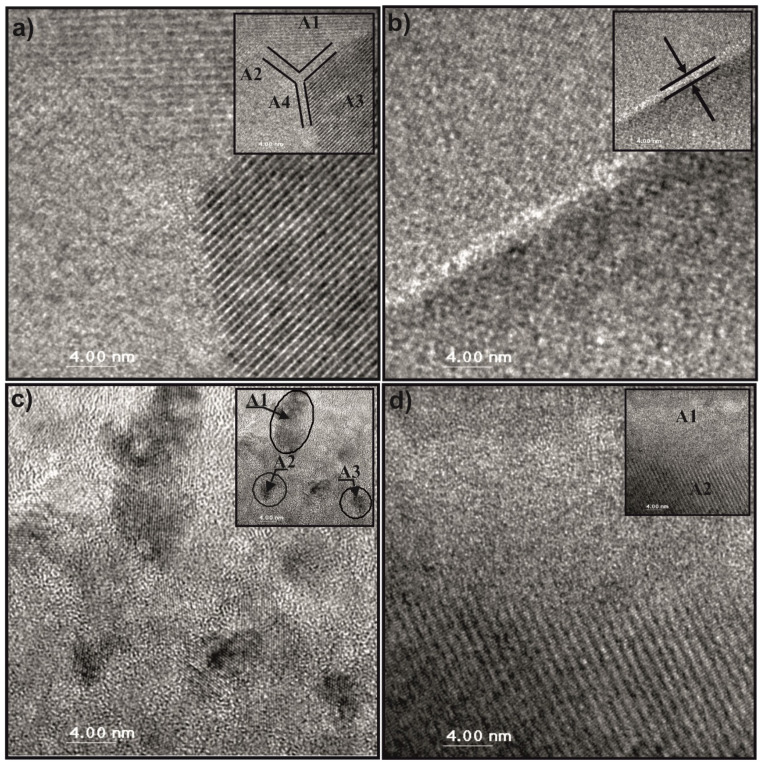
The microstructure of the bonded magnet based on the Nd_14.8_Fe_76_Co_4.8_B_4.2_Nb_0.2_ compound, obtained by a high-resolution transmission electron microscope with highlighted areas of (**a**) four neighboring grains, (**b**) amorphous borders between grains, (**c**) partially crystallized areas, (**d**) amorphous inclusion.

**Figure 10 materials-14-02849-f010:**
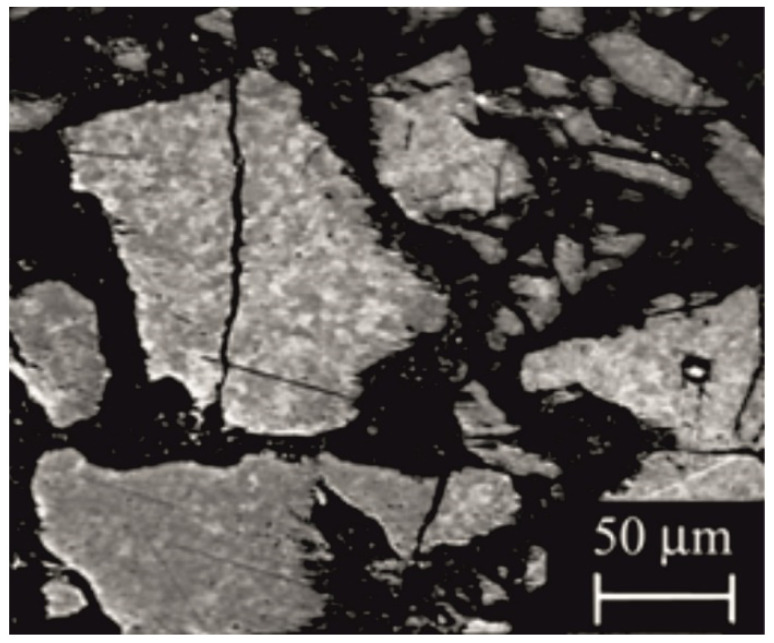
Domain structure of non-magnetized sample, received by the magneto-optical Kerr effect using the Neophot 32 metallographic microscope.

**Figure 11 materials-14-02849-f011:**
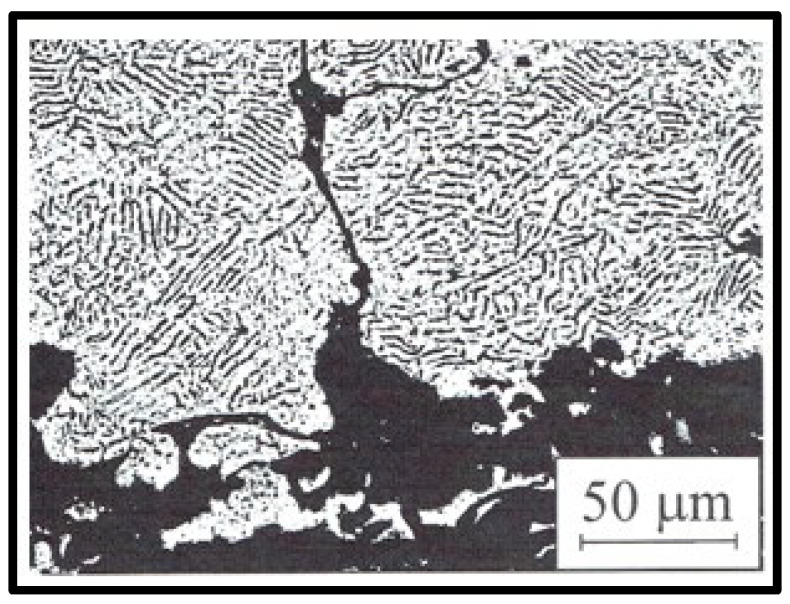
Domain structure of the bonded magnet based on the Nd_14.8_Fe_76_Co_4.8_B_4.2_Nb_0.2_ compound, obtained by the Bitter powder figures method using the Neophot 32 metallographic microscope.

**Figure 12 materials-14-02849-f012:**
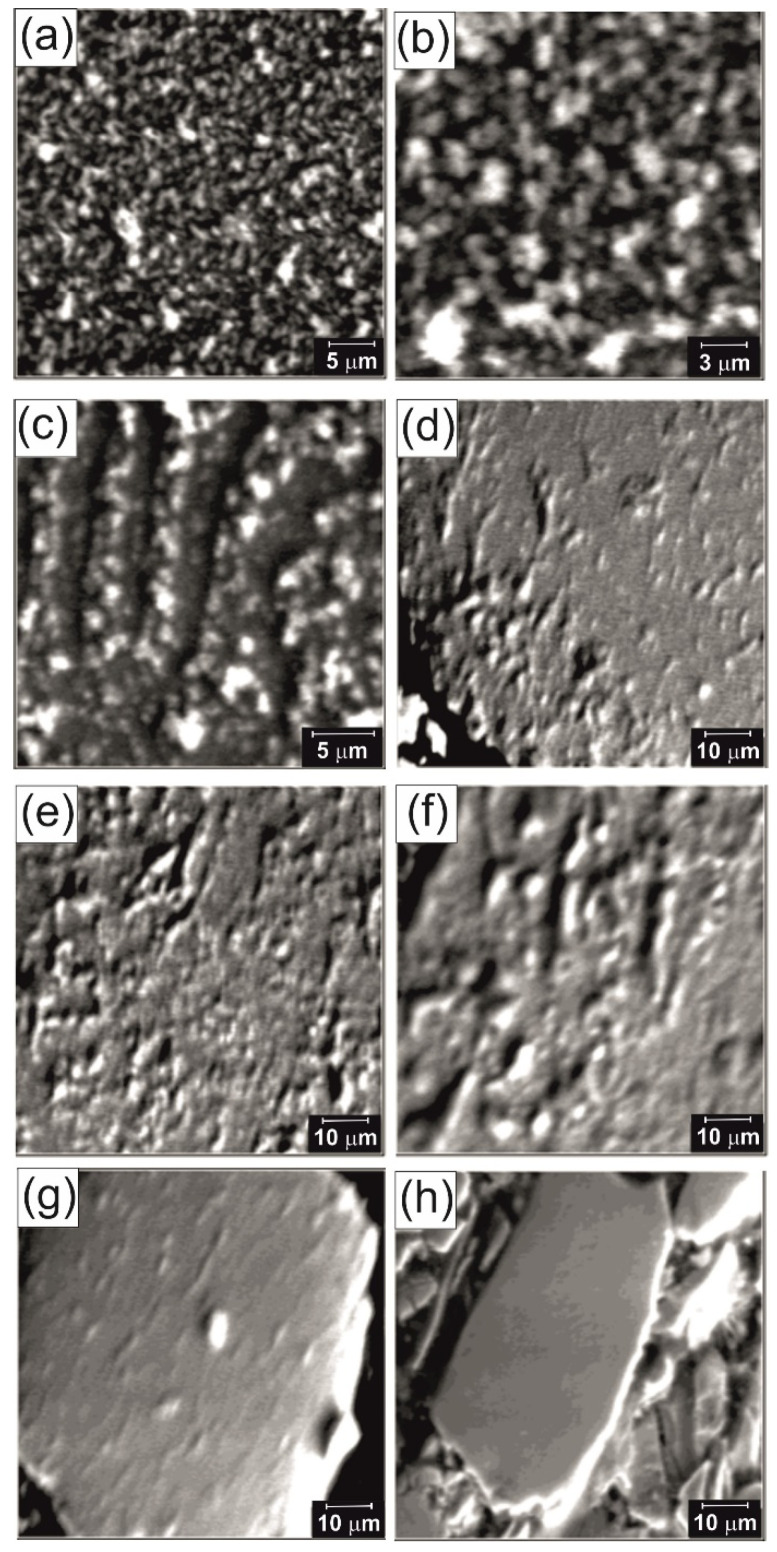
The surface domain structure of the sample: (**a**) not magnetized, magnification ×6000, (**b**) same section area, magnification ×12000, (**c**) of the domain in remanent state after magnetization of 0.6 T field, (**d**–**g**) after magnetization of 0.9 T field, (**h**) lack of domains in remanent state after magnetization of 1.2 T field.

**Figure 13 materials-14-02849-f013:**
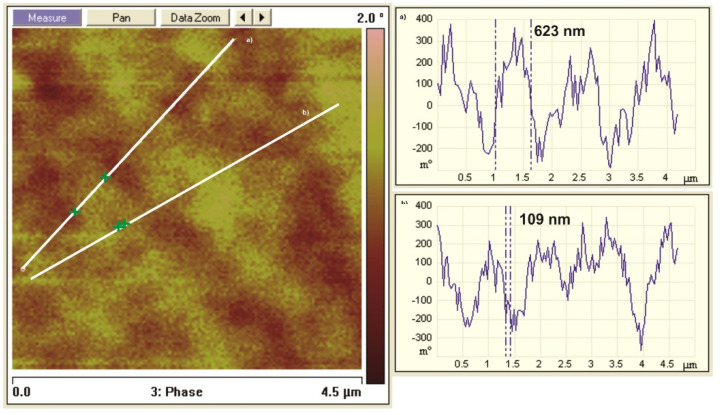
*The MFM domain structure for the sample in the remanence state, with two chosen oscillation profiles marked across line (**a**) and (**b**).*

**Table 1 materials-14-02849-t001:** Fitting parameters of Mössbauer spectra for magnets made from Nd_14.8_Fe_76_Co_4.8_B_4.2_Nb_0.2_ powder.

Fitting Parameter/Position	4c	4e	8j_1_	8j_2_	16k_1_	16k_2_	Paramagnetic Doublet
**Line**	1	2	3	4	5	6	7
**Isomeric shift** **IS (mm/s)**	0.12	0.06	0.06	−0.08	0.02	0.21	−0.13
	±0.02	±0.01	±0.01	±0.01	±0.00	±0.00	±0.02
**Quadrupole splitting** **QS (mm/s)**	0.75	−0.12	−0.20	−0.59	−0.34	−0.11	−0.57
	±0.05	±0.03	±0.02	±0.02	±0.01	±0.01	±0.03
**Induction of hyperfine field μ_0_H_hyp_ (T)**	28.31	25.19	29.12	33.52	28.50	29.23	-
	±0.19	±0.15	±0.08	±0.08	±0.04	±0.02	

**Table 3 materials-14-02849-t003:** Experimental d_hkl_ distances between planes determined from electronograms obtained for the grains shown in Figure 8 in comparison with theoretical data [22].

hkl	$\left(\bar{1}0\bar{1}\right)$	$\left(\bar{1}0\bar{3}\right)$	(002)	$\left(10\bar{1}\right)$
d_hkl_ experimental spacing	7.140	3.729	6.145	7.025
d_hkl_ theoretical spacing [22]	7.140	3.693	6.102	7.140

**Table 2 materials-14-02849-t002:** Quantitative phase composition evaluated from Mössbauer spectra analysis.

Phase Composition	Fe Site	Relative Intensity (%)
**Nd_2_Fe_14_B**	4e	7.01
	4c	7.01
	8j_1_	14.01
	8j_2_	14.01
	16k_1_	28.03
	16k_2_	28.03
	total intensity (%)	98.10
	total intensity error (%)	0.04
**Nd_1.1_Fe_4_B_4_**	total intensity (%)	1.90
	total intensity terror (%)	0.12

## Data Availability

All data used in the paper are archived in the Department of Physics, Faculty of Production Engineering and Materials Technology, Czestochowa University of Technology.

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
