# Peer review of "The Domain and Microstructure of Resin-Bonded Magnets"

_materials, 2021, doi:10.3390/ma14112849_

Round 1

Reviewer 1 Report

The paper presents the domain and structure studies of bonded magnets made from the nanocrystalline powder. It offers a very complex comparison of the methods based on the scanning electron microscopy, high resolution TEM microscopy, Mössbauer spectroscopy and X-ray diffractometry. However, I have some comments and recommendations for the author.

  1. From the article it is not clear, what is the contribution of the article, what is the original part of an author
  2. It is not common to use references for the first time in the conclusion section. This part should summarize the original results presented in the article.
  3. English language has to be revised. The terminology is ok, but articles are missing.
  4. The formal part of the article has to be improved (typography – such for example italics in the formulas, variables etc., work with the template…).

In my opinion the most valuable (and original) part of the article is in its complexity. Usually, the article deals only with a specific methodology. In the article, a very complex comparison of the methods based on the scanning electron microscopy, high resolution TEM microscopy, Mössbauer spectroscopy and X-ray diffractometry is described. The performed overview of methods, which are clearly described and supplemented by the original research, will be in my opinion very interesting for the readers. Despite of the grammar mistakes (missing articles), the text is clear and easy to read. I only wanted from the author to exclude references from the conclusion and to emphasize the importance (and originality) of the achieved results also based on the previous work. 

Author Response

The response to the Reviewer's remarks is included in the attachment.

Reviewer 2 Report

Dear author,

Thank you very much for your interesting paper. I think it is interesting to read and gives insights into microstructural and domain structure of NdFeB magnets. Still, the manuscript can be strongly improved, including a substantial improvement regarding the language. Here are some comments

  • moderate grain size, "mean grain size" might fit better (whereever applicable)
  • Line 43: You asked for a "fine" microstructure. What is fine?
  • Regarding the last part of the introduction, you tell that no exchange of TM elements takes place in your work. Thus, it's not relevant for the introduction. However, the material under investigation contains some cobalt. Consequently, it might be interesting to discuss this topic!
  • Figure 3: Please provide a figure of better quality
  • chapter 3.2. please provde method (e.g. line interception method) for particle size determination
  • as you describe the log-normal distribution as the best fit, which other fits were tested by you?
  • The TEM-technique (equipment) to measure the chemical compositon is missing in the experimental part of the manuscript.
  • Line 237: I cannot follow the argument of setting the composition of B to be 5.8at% in your material containing (nominally) 4.2% B
  • It might help the reader when you show the direction of previously applied field in Figure 12 and Figure 13
  • Figure 13: please provide previously applied magnetic field in figure caption
  • L365: "The moderate grain size and was smaller..." There might be something missing.

Some typos, grammar mistakes:

Line 19: were -> where

L30: "are required" at the end of sentence

L36 tree routs -> three routes

L69: remove "each other"

L78: 180°C

L84: I guess, it is a "Bruker"

L84: please provide type of fiffractometer

L133: have -> has

wherever applicable: wasn't -> was not, didn't -> did not

Table 2: Check it, something from the original document is still there

Table 3: caption is written twice

microns ->µm (whereever applicable)

L360: allowed -> allow

There might be a couple more, please thoroughly check your manuscript again.

Author Response

The response to the Reviewer's remarks is included in the attachment. Unfortunately system do not allow to attach paper with performed changes, but only answer.

Best regards, MD 

Reviewer 3 Report

In this manuscript, the authors present domain and structure studies of bonded magnets made from nanocrystalline Nd-(Fe, Co)-B powder. They used scanning electron microscopy, high-resolution TEM microscopy, Mössbauer spectroscopy, and X-ray diffractometer for structure studies. The authors found labyrinth shape domain structure, typically observed in magnets existed strong exchange interactions between grains. The overall manuscript looks nice and worth publishing but there remain several points to be considered as follows. 

Comment:

  1. The authors should correct the typo error on page no. 3 line no. 78.

Author Response

The response to the Reviewer's remarks is included in the attachment. Unfortunately system do not allow to attach paper with performed changes, but only answer.

Round 2

Reviewer 2 Report

Thank you very much for providing the changes.